# Surprising Instabilities in Training Deep Networks and a Theoretical Analysis

**Yuxin Sun**[1]    **Dong Lao**[2]    **Ganesh Sundaramoorthi**[3]    **Anthony Yezzi**[1]

[1]Georgia Institute of Technology,    [2]UCLA,    [3]Raytheon Technologies

{syuxin3,ayezzi}@gatech.edu, lao@cs.ucla.edu, ganesh.sundaramoorthi@rtx.com

## Abstract

We discover restrained numerical instabilities in current training practices of deep networks with stochastic gradient descent (SGD). We show numerical error (on the order of the smallest floating point bit) induced from floating point arithmetic in training deep nets can be amplified significantly and result in significant test accuracy variance, comparable to the test accuracy variance due to stochasticity in SGD. We show how this is likely traced to instabilities of the optimization dynamics that are restrained, i.e., localized over iterations and regions of the weight tensor space. We do this by presenting a theoretical framework using numerical analysis of partial differential equations (PDE), and analyzing the gradient descent PDE of a simplified convolutional neural network (CNN). We show that it is stable only under certain conditions on the learning rate and weight decay. We reproduce the localized instabilities in the PDE for the simplified network, which arise when the conditions are violated.

## 1   Introduction

Deep learning is now standard practice in a number of application areas including computer vision and natural language processing. However, the practice has out-paced the theory. Theoretical advances in deep learning could improve practice, and could also speed the adoption of the technology in e.g., a number of safety critical application areas.

In this paper, we discover a phenomena of *restrained instabilities* in current deep learning training optimization. Numerical instabilities are present but are restrained so that there is not full global divergence, but these restrained instabilities still cause tiny numerical finite precision errors to amplify, which leads to significant variance in the test accuracy. We introduce a theoretical framework using tools from numerical partial differential equations (PDE) to provide insight into the phenomena and analytically show how such restrained instabilities can arise as a function of learning rate and weight decay for a simplified CNN model. This study is a step toward a theory for principally choosing learning rates in deep network training from the perspective of numerical conditioning.

In deep learning practice, it is well known that large learning rates can cause divergence and small learning rates can be slow and cause the optimization to be stuck at high training error [1], [2]. Learning rates and schedules are often chosen heuristically to address this trade-off. On the other hand, in numerical PDEs, there is a mature theory for choosing the learning rate [3] or step size for discrete schemes for solving PDEs. There are known conditions on the step size (e.g., the Courant-Friedrichs-Lewy (CFL) condition [4]) to ensure that the resulting scheme is stable and converges. These methods would thus be natural to study learning rate schemes for training deep networks with SGD, which in the continuum limit is a PDE. However, to the best of our knowledge, numerical PDEs have not been applied to this domain. This might be for two reasons. First, the complicated structure of deep networks with many layers and non-linearities makes it difficult to apply these analytic techniques. Second, the stochastic element of SGD complicates applying these techniques,

36th Conference on Neural Information Processing Systems (NeurIPS 2022).

which primarily do not consider stochastic elements. While we do not yet claim to overcome these issues, we do show the relevance of PDE analysis in analyzing training of deep networks with SGD, which gives promise for further developing this area.

Our particular contributions are: **1.** We found convincing evidence of and thus discovered the presence of restrained numerical instabilities in standard training practices for deep neural networks. **2.** We show that this results in inherent floating points arithmetic errors having divergence effects on the training optimization that are even comparable to the divergence due to stochastic variability in SGD. **3.** We present and study a simplified model, which allows us to exploit numerical PDE analysis, in order to offer an explanation on how these instabilities might arise. [1]

These insights suggest to us that the learning rate should be chosen adaptively over localized space-time regions of the weight tensor space in order to ensure stability while not compromising speed. While we understand this is a complex task, without a simple solution, our study motivates the need for investigating such challenging strategies. Note there are a number of heuristically driven methods for adaptive learning rates (e.g., [5]–[9]), but they do not address numerical stability.

## 2   Related Work

As stated earlier, we are not aware of related work in applying numerical PDE analysis to study stability and choice of learning rates of deep network training [2]. There has been work on studying convergence of SGD and learning rates (e.g., [10]–[12]), but they are often based on convexity and Lipschitz assumptions that could be difficult to apply to deep networks. For deep networks, there is work on global stability of training in relation to curvature of the loss, e.g., [13]. [13] is more focused on generalization rather than numerical stability and does not address restrained instabilites.

More broadly, although not directly related to our work, there has been a number of recent works in connecting PDEs with neural networks for various end goals (see [14]–[16] for detailed surveys). For example, applying neural networks to solve PDEs [17]–[21], and interpreting the forward pass of neural networks as approximations to PDEs [22]–[24]. Since SGD can be interpreted as a discretization of a PDE, PDEs also play a role in developing optimization schemes for neural networks [25]–[31].

Despite various methods [32]–[34] proposed to reduce variance of final accuracy induced by the stochastic gradient, we find quite surprisingly that even if all random seeds in optimization are fixed, variance induced by epsilon-small numerical noise (perturbing only the last floating point bit) and hence floating point arithmetic is nevertheless greater or similar to variance across independent trials (see Table 1). This observation potentially indicates that numerical stability is a critical but long-ignored factor affecting neural network optimization.

There has been work on training deep networks using limited numerical precision (e.g., half precision) [35], [36]. The purpose of these works is to introduce schemes to retain accuracy, which is well known to be reduced significantly when lowering precision to these levels. Different from this literature, we show an unknown fact: errors just in the last bit of floating point representations result in significant divergence. This is to provide evidence of numerical instability resulting from learning rate choices.

## 3   Background: PDEs, Discretization, and Numerical Stability

We start by providing background on analyzing stability of linear PDE discretizations. This methodology will be applied to neural networks in later sections as a PDE is the continuum limit of SGD. We illustrate concepts using a basic PDE, i.e., the heat equation (used to model a diffusion process):

$$\partial_t u(t, x) = \partial_{xx} u(t, x), \quad u(0, x) = u_0(x), \tag{1}$$

where $u : [0, T] \times \mathbb{R} \to \mathbb{R}$, $\partial_t u$ denotes the partial with respect to time $t$, $\partial_{xx} u$ is the second derivative with respect to the spatial dimension $x$, and $u_0 : \mathbb{R} \to \mathbb{R}$ is the initial condition. To solve

---

[1]While this first step to study deep networks using more matured tools for the numerical analysis of discretized PDE's is focused on a 1-layer CNN, we show in the appendix how this starting point still yields useful insight into the more complicated dynamics of multi-layer networks, especially when highly regularized, thereby offering strong promise for further developing this theory.

[2]In deep learning, "stability" often refers to vanishing/exploding gradients, network dynamics, or model robustness. These topics are different from numerical stability of the training optimization studied in this paper.

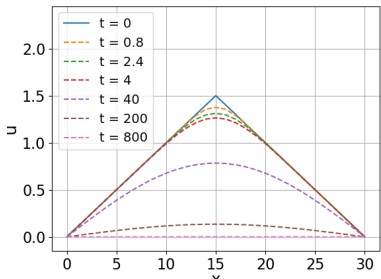 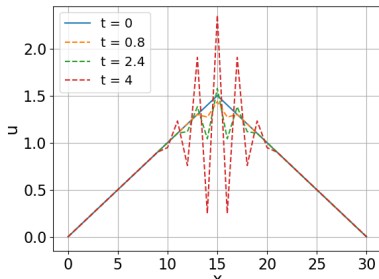

Figure 1: Illustration of instability in discretizing the heat equation. The initial condition $u_0$ is a triangle (blue), with boundary conditions $u(0) = u(30) = 0$. **[Left]**: When the CFL condition is met, i.e., $\Delta t/(\Delta x)^2 = 0.4 < \frac{1}{2}$, the method is stable and approximates the solution of the PDE. Note the true steady state is 0, which matches the plot. **[Right]**: When the CFL condition is not met, i.e., $\Delta t/(\Delta x)^2 = 0.8 > \frac{1}{2}$, small numerical errors are amplified and the scheme diverges.

this numerically, one discretizes the equation, using finite difference approximations. A standard discretization is through $\partial_{xx}u(t,x) = \frac{u(t,x+\Delta x) - 2u(t,x) + u(t,x-\Delta x)}{(\Delta x)^2} + O((\Delta x)^2)$ and $\partial_t u(t,x) = \frac{u(t+\Delta t,x) - u(t,x)}{\Delta t} + O(\Delta t)$, where $\Delta x$ is the spatial increment and $\Delta t$ is the step size, which gives the following update scheme:

$$u^{n+1}(x) = u^n(x) + \frac{\Delta t}{(\Delta x)^2} \cdot [u^n(x + \Delta x) - 2u^n(x) + u^n(x - \Delta x)] + \epsilon^n(x), \qquad (2)$$

where $n$ denotes the iteration number, $u^n(x)$ is the approximation of $u$ at $t = n\Delta t$, and $\epsilon$ denotes the error of the approximation (i.e., due to discretization and finite precision arithmetic).

A key question is whether $u^n$ remains bounded as $n \to \infty$, i.e., *numerically stable*. It is typically easier to understand stability in the frequency domain, which is referred to as Von Neumann analysis [3], [37]. Computing the spatial Discrete Fourier Transform (DFT) yields:

$$\hat{u}^{n+1}(\omega) = A(\omega)\hat{u}^n(\omega) + \hat{\epsilon}^n(\omega), \quad \text{or} \quad \hat{u}^n(\omega) = A^n(\omega)\hat{u}_o(\omega) + \sum_{i=0}^{n-1} A^i(\omega)\hat{\epsilon}^{n-i}(\omega), \qquad (3)$$

where the hat denotes DFT, $\omega$ denotes frequency, and $A(\omega) = 1 - \frac{\Delta t}{(\Delta x)^2}[1 - \cos(\omega\Delta x)]$ is the amplifier function. Note that $u^n$ is stable if and only if $|A(\omega)| < 1$. Notice that if $|A(\omega)| < 1$, $u^n$ converges and errors $\epsilon^n$ are attenuated over iterations. If $|A(\omega)| \geq 1$, $u^n$ diverges and the errors $\epsilon^n$ are amplified. The case, $|A(\omega)| < 1$ for all $\omega$, implies the conditions that $\Delta t > 0$ and $\Delta t < \frac{1}{2}(\Delta x)^2$. This is known as the CFL condition. Notice the restriction on the time step (learning rate) for numerical stability. If the discretization error of the operator on the right hand side of the PDE is less than $O(\Delta x)$, stability also implies convergence to the PDE solution as $\Delta t \to 0$ [3], [37].

See Figure 1 for simulation of the discretization scheme for the heat PDE. The instability causes blowup of the solution globally. For standard optimization of CNNs, we will show rather that the instability is *restrained* (localized in time and space), which nevertheless causes divergence.

## 4 Empirical Evidence of Instability in Training Deep Networks

We discover and provide empirical evidence of restrained instabilities in current deep learning training practice. To this end, we show that optimization paths in current practice of training convolutional neural networks (CNNs) can diverge significantly due to the smallest errors from finite precision arithmetic. We show that the divergence can be eliminated with learning rate choice.

### 4.1 A Perturbed SGD by Introducing the Smallest Floating Point Perturbations

We introduce a modified version of SGD that introduces perturbations of the original SGD update on the order of the smallest possible machine representable perturbation, modeling noise on the order of

error due to finite precision floating point arithmetic:

$$\theta_{t+1} = \theta_t - \eta g_t \hspace{4cm} \text{(SGD)}$$
$$\theta_{t+1} = \theta_t - (\eta/k) \times (kg_t), \hspace{3cm} \text{(PERTURBED SGD)}$$

where $\eta > 0$ is the learning rate, $g_t$ is the stochastic gradient or the momentum vector if momentum is used, and $k > 1$ is an odd integer. Notice that mathematically, both updates are exactly equivalent (division by $k$ cancels the multiplication by $k$), however considering floating point arithmetic, these may not be equivalent. A floating point number is represented as

$$\text{significand} \times \text{base}^{\text{exponent}} \hspace{0.5cm} (s \text{ bits significand}, e \text{ bits exponent}) \hspace{2cm} (4)$$

where typically base 2 (binary) is used. Thus, the modified update introduces a potential perturbation in the last significant bit of $\eta g_t$, hence a relative difference of the right hand sides on the order of $2^{-s}$ according to IEEE standards on floating point computation. In other words, $\text{RelativeError}(x, y) := |x - y|/|x| < 2^{-(s-1)}$ where $x = \text{fl}(\eta g_t)$, $y = \text{fl}(\eta/k) \times \text{fl}(kg_t)$, and $\text{fl}(x)$ is the floating point representation of $x$. Note that this is the same order of relative error between $\eta g_t$ and $\text{fl}(\eta g_t)$, the inherent machine error due to floating point representation. The perturbation induces the *smallest* possible perturbation of $\eta g_t$ representable by the machine. Thus, perturbed SGD is a way to show the effects of machine error. Notice that when $k$ is a power of two, as floating point numbers are typically stored in binary form, the division just subtracts the exponent and the multiplication adds to the exponent so both updates are equivalent and no perturbation is introduced. However, choosing $k$ odd can create changes at the last significant bit, with each choice of $k$ yielding a different change.

### 4.2 Perturbed SGD on Common CNNs and Demonstration of Restrained Instabilities

In SGD, there are several sources of randomness, i.e., random initialization, random shuffled batches/selection, random data augmentation, and there are also sources of non-determinism in implementations of the deep learning frameworks. To ensure the variance merely originated from the floating point perturbation, in the following experiments, we eliminate all randomness by fixing the initialization, the random seed for batch selection, and the non-determinism flag. Appendix B empircally validates that all seeds are fixed. We use Pytorch using default 32 bit floating point precision (similar conclusions for 64 bit hold).

**Perturbed SGD with ReLU**: Our first experiment uses the ResNet-56 architecture [38], which we train on CIFAR-10 [39] using perturbed SGD. We use the standard parameters for training this network [38]: `lr=0.1, batch size = 128, weight decay = 5e-4, momentum=0.9`. The standard step decay learning rate schedule is used: the learning rate is divided by 10 every 40 epochs for a total of 200 epochs. We report the final test accuracies for 6 different values of $k$, including $k = 1$ (standard SGD), and compute the standard deviation of the accuracies (STD). Table 1 (left) shows the results. This results in test accuracy variability over different $k$. We repeat this experiment over different seeds for the stochastic batch selection (shown in the next columns) fixing the initialization. The standard deviation for each seed over $k$ is on average $0.16\%$. To illustrate the significance of this value, we compare to the variability due to batch selection on the right of Table 1. We empirically estimate the relative fluctuation of the gradient estimate from batch selection, which is $26.72$. This is large compared to $2^{-23}$ for the floating point perturbation, yet this smallest fluctuation that is machine representable (and models finite precision error) results in the about the same accuracy variance as batch selection. Thus, finite precision error, inherent in the system, surprisingly results in about the same test accuracy variance as stochastic batch selection.

**Perturbed SGD with Swish**: We first conjectured that the variance due to floating point errors could be caused by the ReLU activation, the activation standard in ResNets. This because the gradient of the ReLU is discontinuous at zero. Hence small positive values (e.g., 1e-6) would give a gradient of 1 and small negative values (e.g., -1e-6) would give a gradient of zero. Hence the ReLU gradient is sensitive to numerical fluctuation around zero. Note common deep learning implementations choose a value at zero, among 0, 0.5 or 1; but this has the same sensitivity. Therefore, we adopted the Swish activation function (proposed in [40]), $\text{Swish}(x) := x/[1 + \exp(-\beta x)]$ ($\beta > 0$, as $\beta \to \infty$ Swish approaches ReLU), which is is an approximation of the ReLU that has a continuous gradient at zero. Recent work has shown the Swish function improves accuracy [40] as well. Thus, we repeated the previous experiment (under the same settings) replacing the ReLU with the Swish activation. Results are shown in Appendix, and show similar results as the previous experiment with ReLU: the

| SEED | 1 | 2 | 3 | 4 | 5 | 6 | STD |
|------|------|------|------|------|------|------|------|
| $k = 1$ | 93.36 | 93.40 | 93.10 | 93.14 | 93.34 | 93.33 | 0.11 |
| $k = 3$ | 93.49 | 93.37 | 93.08 | 93.68 | 93.16 | 93.12 | 0.22 |
| $k = 5$ | 93.64 | 93.22 | 93.39 | 93.17 | 93.26 | 93.42 | 0.16 |
| $k = 7$ | 93.36 | 93.31 | 93.12 | 93.23 | 93.14 | 93.28 | 0.09 |
| $k = 9$ | 93.87 | 93.55 | 93.08 | 93.35 | 93.42 | 93.41 | 0.24 |
| $k = 11$ | 92.99 | 93.31 | 93.49 | 93.48 | 93.14 | 93.56 | 0.21 |
| STD | 0.27 | 0.10 | 0.16 | 0.19 | 0.10 | 0.13 | |

| PERTURBATION METHOD | RELATIVE GRAD. ERROR | TEST ACC STD |
|---------------------|----------------------|--------------|
| FLOATING PT | $2^{-23}$ | 0.16 |
| BATCH SELECTION | 26.72 | 0.17 |

Table 1: **[Left]**: Final test accuracy over different seeds (batch selections) and different floating point perturbations (rows) for Resnet56 trained on CIFAR-10. STD is standard deviation. **[Right]**: Comparison of errors induced in the gradient and resulting test accuracy STD for floating point perturbation vs batch selection. Floating point error perturbs the gradient by an amount 8 orders of magnitude smaller than batch selection yet surprisingly yields nearly the same test accuracy variation!

standard deviation in test accuracy due to floating point perturbation ($93.72 \pm 0.15$) is significant and is comparable (even exceeds) the variance due to stochastic batch selection (0.09).

**Other Architectures/Datasets**: To verify that this phenomenon is not specific to the choice of network architecture or dataset, we repeated the same experiment for a different network (VGG16 [41]) and a different dataset (Fashion-MNIST [42]). We summarize the results in the Appendix, which confirms that the phenomena is still present across different networks and datasets.

**Divergence in Optimization Paths**: We plot the difference in network weights (using the measure described next) between the original SGD weights, $\theta_i$ ($i$ is the index for location in the vector), and the perturbed SGD weights, $\theta'_i$ over epochs and show that the errors are amplified, suggesting instability. We use the average relative L1 absolute difference between weights: $\text{RelL1}(\theta, \theta') := \frac{2}{N} \sum_i |\theta_i - \theta'_i| / (|\theta_i| + |\theta'_i|)$, where $N$ is the number of network parameters. This measure is used as the numerical error is multiplicative and relative to the weight. Note that RelL1 is bounded by 2. Results are shown for multiple networks in Figure 2 (left). The difference rapidly grows, indicating errors are amplified, and then the growth is restrained at larger epochs. Although the errors are restrained, the initial error growth is enough to result in different test accuracies.

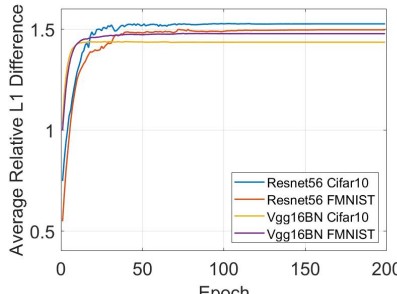 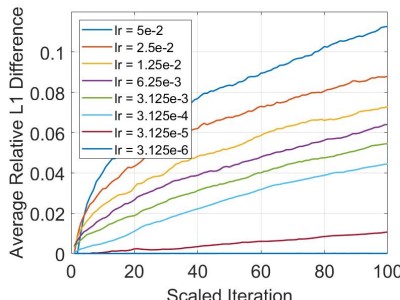

Figure 2: **[Left]**: Relative L1 difference in weights across epochs for SGD ($k = 1$) and modified SGD ($k = 3$). A decayed learning rate schedule is used. The errors quickly build up, but then are contained at higher epochs. The initial build up of errors is enough to result in different test accuracy. **[Right]**: Relative L1 difference in weights over initial iterations for SGD ($k = 1$) and modified SGD ($k = 3$) with various fixed learning rates. Lower than some minimum learning rate, the floating point perturbation gets attenuated and higher than that value errors are amplified, suggesting an instability.

**Evidence Divergence is Due to Instability**: We provide evidence that this divergence phenomenon is due to an instability by showing that decreasing the learning rate to a small enough value can eliminate the divergence, even initially. We experiment with various fixed learning rates. We use Resnet56 under the same settings as before. For smaller learning rates, more iterations would have to be run to move an equivalent amount for a larger rate. We thus choose the iterations for each run such that the learning rate times the number of iterations is constant. The result is shown in Figure 2 (right), where the axis is scaled as mentioned. When the learning rate is chosen small enough, the

floating point errors are attenuated over iterations, indicating the process is stable (note it is not stuck at a local minima - see Appendix C). Above this level, the errors are amplified.

# 5 PDE Stability Analysis of a Simplified CNN

In this section, we use numerical PDE tools introduced in Section 3 to analyze a one-layer CNN to show analytically how the instability phenomenon described in the previous section can arise. To do this, we start with a continuum model of the network, derive the analytical formula for the gradient descent, which is a PDE, discretize it, which serves as a model for the optimization algorithm, and analyze the stability of the discretization through Von-Neumann analysis.

## 5.1 Network, Loss and Gradient Flow PDE

Let $I : \mathbb{R}^2 \to \mathbb{R}$ be an input image to the network, $K : \mathbb{R}^2 \to \mathbb{R}$ be a convolution kernel (we assume $I$ and $K$ to be of finite support), $r : \mathbb{R} \to \mathbb{R}$ be the Swish activation, and $s : \mathbb{R} \to \mathbb{R}$ is the sigmoid function. We consider the following CNN, which is the simplest version of VGG:

$$f(I) = s \left[ \int_{\mathbb{R}^2} r(K * I)(x) \, \mathrm{d}x \right]. \tag{5}$$

This network consists of a convolution layer, an activation layer, a global pooling layer and a sigmoid layer. As in deep learning packages, $*$ will denote the cross-correlation (which we call convolution). The bias is not included in the network as it does not impact our analysis nor conclusions.

We assume a dataset that consists of just a single image $I$ and its label $y \in \{0, 1\}$. This is enough to demonstrate the instability phenomena. We consider the following regularized cross-entropy loss:

$$L(K) = \ell(y, \hat{y}) + \frac{1}{2} \alpha \|K\|_{\mathbb{L}^2}^2, \quad \text{where} \quad \hat{y} = f(I), \tag{6}$$

$\ell$ denotes cross-entropy, the second term is regularization ($\mathbb{L}^2$ norm squared of the kernel), and $\alpha > 0$ is a parameter (weight decay). Conclusions are not are restricted to cross-entropy, but it makes some expressions simpler. The gradient descent PDE is given by (see the Appendix for proof):

**Theorem 5.1** (Gradient Descent PDE). *The gradient descent PDE with respect to the loss* (6) *is*

$$\partial_t K = -\nabla_K L(K) = -(\hat{y} - y) r'(K * I) * I - \alpha K, \tag{7}$$

*where $r'$ denotes the derivative of the activation, and $t$ parametrizes the evolution. If we maintain the constraint that $K$ is finite support so that $K$ is zero outside $[-w/2, w/2]^2$ then the constrained gradient descent is*

$$\partial_t K = [-(\hat{y} - y) r'(K * I) * I - \alpha K] \cdot W, \tag{8}$$

*where $W$ is a windowing function (1 inside $[-w/2, w/2]^2$ and zero outside).*

## 5.2 Stability Analysis of Optimization

We now analyze the stability of the standard discretization of the gradient descent PDEs. Note that the PDEs are non-linear, and in order to perform the stability analysis, we study linearizations of the PDE around the switching regime of the activation, i.e., $K = 0$. As we will see in the next sub-section, away from the transitioning regime, a similar analysis applies.

We first study the linearization of (7), and then we adapt the analysis to (8). The linearization of the PDE results in the following (see Appendix for the derivation):

**Theorem 5.2** (Linearized PDE). *The linearization of the PDE* (8) *around $K = 0$ is given by*

$$\partial_t K = \left[ -\frac{a}{2} (\bar{I} + \beta [(K * I) * I]) - \alpha K \right] W, \quad \text{where} \quad a = s' \partial_{\hat{y}} \ell = \hat{y} - y, \tag{9}$$

*$a$ is considered constant with respect to $K$, $\bar{I}$ denotes the integral (sum) of the values of $I$ (assumed finite support), and $\beta > 0$ is the parameter of the Swish activation, $r(x) := \frac{x}{1 + e^{-\beta x}}$.*

In Appendix D, we also consider a non-constant linear model of $K$ for $a$. This leads to the similar conclusions as the constant model, and so we present the constant model for simplicity.

We discretize the linearization, which results in a discrete optimization algorithm, equivalent to the algorithm employed in standard deep learning packages. Using forward Euler discretization gives

$$K^{n+1} - K^n = \left[-\frac{a}{2}\Delta t(\bar{I} + \beta[(K^n * I) * I]) - \Delta t \alpha K^n\right] W, \tag{10}$$

where $n$ denotes the iteration number, and $\Delta t$ denotes the step size (learning rate in SGD). To derive the stability conditions, we compute the spatial DFT of (10) (see the Appendix D for details):

**Theorem 5.3** (DFT of Discretization). *The DFT of the discretization* (10) *of the linearized PDE is*

$$\hat{K}^{n+1} - \hat{K}^n = -\frac{a}{2}\Delta t(\bar{I} + \beta\hat{K}^n|\hat{I}|^2) * \text{sinc}\left(\frac{w}{2}\cdot\right) - \Delta t \alpha \hat{K}^n, \tag{11}$$

*where $\hat{K}^n$ denotes the DFT of $K^n$, and* sinc *denotes the sinc function. In the case that $w \to \infty$ (the window support becomes large), the DFT of $K$ can be written in terms of the amplifier $A$ as*

$$\hat{K}^{n+1}(\omega) = A(\omega)\hat{K}^n(\omega) - \frac{a}{2}\Delta t\bar{I}, \quad \text{where} \quad A(\omega) = 1 - \Delta t\left(\alpha + \frac{1}{2}a\beta|\hat{I}(\omega)|^2\right). \tag{12}$$

In order for the updates to be a stable process, the magnitude of the amplifier should be less than one, i.e., $|A| < 1$. This results in the following stability criteria (see the Appendix D for a derivation):

**Theorem 5.4** (Stability Conditions). *The discretization of the linearized PDE* (7) *whose DFT is given by* (11) *is stable (in the case $w \to \infty$, i.e., the window gets large) if and only if the following conditions on the weight decay $\alpha$ and the step size (learning rate) $\Delta t$ are met:*

$$\alpha < \alpha_{max} := \frac{2}{\Delta t} - \frac{1}{2}a\beta \min_{\omega}|\hat{I}(\omega)|^2 \quad \text{and} \quad \alpha > \alpha_{min} := -\frac{1}{2}a\beta \max_{\omega}|\hat{I}(\omega)|^2. \tag{13}$$

Note the sign of $a = \hat{y} - y$ can be either positive or negative; in the case $a > 0$, the second condition is automatically met (since $\beta > 0$). However in the case that $a < 0$, the weight decay must be chosen large enough to be stable. The first condition shows that the weight decay and the step size satisfy a relationship if the optimization is to be stable. Note that the conditions also depend on the structure of the network, the current state of the network (i.e., through dependence of $a$), and the frequency content of the input. This is different than current deep learning practice, where the weight decay is typically chosen constant through the evolution and does not adapt with the state of the network, and the step size is typically chosen empirically without dependence on the weight decay.

## 5.3 Empirical Validation of Stability Bounds for Linear PDE

Figure 3 empirically validates the existence of upper and lower bounds on $\alpha$. We chose the learning rate $\Delta t = $ 1e-8, $a = -0.5, \beta = 1$, initialize the weights $K$ ($32 \times 32$) according to a normal distribution (mean 0, variance 1), and $I$ is a $256 \times 256$ checkerboard pattern filled with 1 and -1.

The above stability conditions were under the condition that the windowing function, $W$, had infinite support. Thus, the evolution could cause the initial kernel's (finite) support to grow over iterations, which is not representative of fixed, finite support kernels used in deep network practice. In the case that $W$ is finite and fixed support, it is not possible to analytically find a simple multiplicative amplifier factor as in (12) since the convolution with the sinc function in (11) does not separate from $\hat{K}^n$. In the case that $\hat{K}$ is a constant (i.e., $K$ is a delta function or support of size 1), $(\hat{K}^n|\hat{I}|^2) * \text{sinc}\left(\frac{w}{2}\cdot\right) = \hat{K}^n(|\hat{I}|^2 * \text{sinc}\left(\frac{w}{2}\cdot\right))$, in which case one can write the amplifier function as

$$A(\omega) = 1 - \Delta t\left(\alpha + \frac{1}{2}a\beta|\hat{I}(\omega)|^2 * \text{sinc}\left(\frac{w}{2}\omega\right)\right), \tag{14}$$

which has the effect that the upper bound for $\alpha$ increases and the lower bound decreases in (13), but maintains existence of upper and lower bounds.

We conjecture this is also true when $K$ has support larger than 1, and the bounds on $\alpha$ converge to (13) as the support of the kernel is increased. We provide empirical evidence. We find the lower and upper bounds for $\alpha$ by running the PDE and noting the values of $\alpha$ when the scheme first becomes

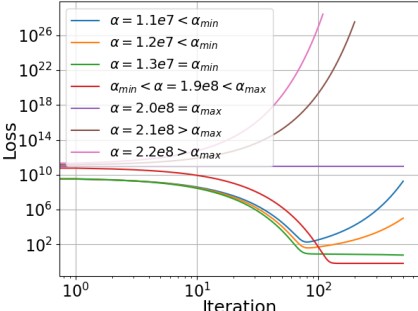

Figure 3: Empirical validation of stability bounds for the linearized PDE (9). When the weight decay is chosen such that $\alpha \in (\alpha_{min}, \alpha_{max})$, the PDE is stable and the loss converges, otherwise the PDE can be unstable and diverge. The pink and brown lines are clipped as the loss exceeded the largest float.

Table 2: Comparison of the bounds on weight decay, $\alpha$, between the non-windowed linear PDE, i.e., $\alpha_{min}$ and $\alpha_{max}$ in (13) and the windowed linear PDE in (9) found empirically, i.e., $\alpha_{min}^e$ and $\alpha_{max}^e$. Verifies bounds exist in windowed case.

| Kernel | $\alpha_{min}^e$ | $\alpha_{min}$ | $\alpha_{max}^e$ | $\alpha_{max}$ |
|--------|--------|--------|--------|--------|
| 16x16 | 3.8e6 | 1.07e9 | 2e8 | 2e8 |
| 32x32 | 1.3e7 | 1.07e9 | 2e8 | 2e8 |
| 64x64 | 3.8e7 | 1.07e9 | 2e8 | 2e8 |

unstable. We compare it to the bounds in (13). We use the same settings as the previous experiment. See results in Table 2, which verifies the bounds approach the infinite support case.

In this section, we analyzed the linearized gradient descent PDE. Momentum is often used. See Appendix E for analysis of the momentum case, which leads to similar conclusions as this section.

### 5.4 Restrained Instabilities in the Non-linear PDE

We now explain how the linear analysis around $K = 0$ in the previous sections applies to analyzing full non-linear PDE. In particular, we show that the non-linear PDE is also not stable unless similar conditions on the weight decay and step size are met. However, the non-linear PDE can transition to regimes that have more generous stability conditions where it could be stable, and the PDE can move back and forth between stable and unstable regimes, resulting in restrained instabilities.

**Approximation of Non-Linear PDE by Linear PDEs and Stability Analysis**: The non-linear PDE could move between three different regimes: when the activation is "activated" ($r'(x) = 1$), "not activated" ($r'(x) = 0$), and "transitioning" ($r'(x)$ is a non-constant linear function). In the transitioning regime, $K \approx 0$, in which case the linear analysis of the previous section applies, and in this regime, there are conditions on the weight decay and step size for stability. In the not-activated regime, $K * I$ is negative and away from zero, the non-linear PDE is driven only by the regularization term, which is stable when $\alpha < \frac{2}{\Delta t}$. This is typically true in current deep learning practice where weight decay is chosen on the order of 1e-4 and the learning rate is on order of <1e-1. In the activated regime, $K * I$ is positive and away from zero and the non-linear PDE can be approximated by a linear PDE (see Appendix D.5), which gives the stability condition: $\alpha < \frac{2}{\Delta t} - \frac{\bar{I}^2}{8}$, similar to the stability condition in the transitioning regime.

**Formation of Restrained Instabilities**: Note that different regions in the kernel domain could each be in different regimes and therefore governed by different linear PDEs discussed earlier. See Table 3. Such regions could then switch between different regimes throughout the evolution of the PDE, and thus the evolution could go between stable and unstable regimes. For example, in the transitioning regime, the instability quickly drives the magnitude of the kernel up (if the stability conditions are not met) and thus the region to a possibly stable (e.g., activated) regime. Data-driven terms could then kick the kernel back into the (unstable) transitioning regime, and this switching between regimes could happen indefinitely. As a result instabilities occur, but can be "restrained" by the activation. The optimization nevertheless amplifies and accumulates small noise in the transitioning regime.

**Empirical Validation**: We now empirically demonstrate that restrained instabilites are present and error amplification occurs in the (non-linear) gradient descent PDE of the one layer CNN even when the dataset consists of a single image. We choose $\alpha = 20, \beta = 1, y = 1$ and the same input image $I$ and kernel $K$ initialization as specified in Section 5.3. Figure 4 (left) shows the loss function over

| | Activated | Transitioning | Not Activated |
|---|---|---|---|
| Region | $\{x : K * I(x) \gg 0\}$ | $\{x : K * I(x) \approx 0\}$ | $\{x : K * I(x) \ll 0\}$ |
| Linear PDE | $\partial_t K = -(s(0) - y)\bar{I}$ $-\frac{1}{8}\bar{I}^2 \bar{K} - \alpha K$ | $\partial_t K = -\frac{a}{2}(\bar{I} + \beta[(K * I) * I])$ $-\alpha K$ | $\partial_t K = -\alpha K$ |
| Stability | $\alpha < \frac{2}{\Delta t} - \frac{\bar{I}^2}{8}$ | $\alpha < \frac{2}{\Delta t} - \frac{1}{2} a\beta \min_\omega |\hat{I}(\omega)|^2$ $\alpha > -\frac{1}{2} a\beta \max_\omega |\hat{I}(\omega)|^2$ | $\alpha < \frac{2}{\Delta t}$ |

Table 3: The non-linear PDE is approximated by three linear PDEs in different regions of the tensor (kernel) space ("activated", "not activated" and "transitioning"). The non-linear PDE can transition between these regions in which different linear PDEs approximate the behavior.

iterations for various learning rate ($\Delta t$) choices, and the right plot shows the evolution of the L1 relative difference between weights of SGD ($k = 1$) and perturbed SGD ($k = 3$) as discussed in Section 4. From the previous discussion and (13), we know that it is necessary that $\Delta t < 0.1$ in order for all regimes to be stable. This is confirmed by the plot on the left: when $\Delta t > 0.1$, the loss blows up. When $\Delta t < 0.03$, which is the empirically determined threshold for the transitioning region, all regimes are stable, and the kernel converges as shown in the plot. When $0.03 < \Delta t \le 0.1$, the PDE transitions between the (unstable) transitioning regime and the stable one (activated), which introduces oscillations. This is consistent with our theory outlined previously. On the plot on the right of Figure 4, we show divergence caused by the floating perturbations introduced in Section 4 for various $\Delta t$. When $0.03 < \Delta t \le 0.1$ (in the restrained instability regime), the error builds up quickly and then levels off. When $\Delta t < 0.03$, errors are not amplified. This is consistent with the behavior of multi-layer networks in Section 4 validating our theory.

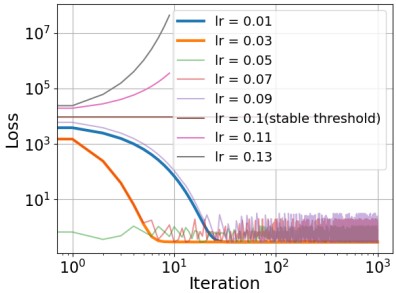
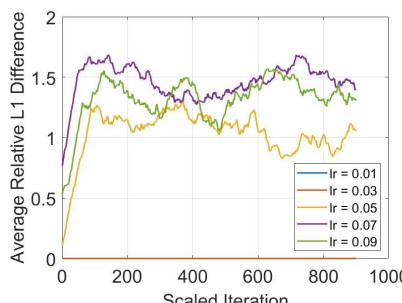

Figure 4: **[Left]**: Loss vs iterations of the non-linear PDE (8) for various choices of learning rates ($\Delta t$). The loss for various learning rates are consistent with the theory predicted for fully stable, fully unstable and restrained instabilities. **[Right]**: L1 error accumulation in the non-linear PDE. The plot is consistent with expected error accumulation for restrained instabilities, and fully stable regimes.

**Remarks on Deeper Networks**: This section formulated theory on how the divergence phenomena for practical deep network optimization reported in Section 4 could arise. We found that even the simplest (one-layer) CNN exhibits this phenomena through analytical techniques. In multi-layer networks, there are more possibilities for such restrained instabilities to arise in multiple layers and multiple kernels of the network. Such instabilities may only arise in some kernels/layers. The analysis in this section is relevant to any one layer of such networks by treating the part of the network before the layer as input; within a small time period, one can treat this an a nearly constant input to the layer.

## 6 Conclusion

We discovered restrained numerical instabilities in standard training procedures of deep networks. In particular, we showed that epsilon-small errors inherent in floating point arithmetic can amplify and lead to divergence in final test accuracies, comparable to stochastic batch selection. Given such variations are comparable to accuracy gains reported in many papers, such instabilities could inflate or even suppress such gains, and further investigation is needed. Using a simplified CNN, which is

amenable to linear analysis, we employed PDEs to explain how these instabilites might arise. We derived CFL conditions, which imposed conditions on the learning rate and weight decay.

This study provided a step in principally choosing learning rates for stability. We showed that numerical PDEs have promise to shed insight. The analysis showed how restrained instabilities can arise even in a simplified CNN, which suggests the same phenomena exists in larger networks. Although empirically we showed how restrained instabilites could be eliminated with globally small learning rates, such instabilities are localized to space-time regions of the tensor space, suggesting in practice the need to develop adaptive learning rate schemes tailored to have small rates in these local regions to ensure numerical stability, while maintaining speed.

## Acknowledgements

This research was supported in part by Army Research Labs (ARL) W911NF-22-1-0267 and Raytheon Technologies Research Center.

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
