# OpenReview forum: "Surprising Instabilities in Training Deep Networks and a Theoretical Analysis "
_NeurIPS.cc/2022/Conference — NeurIPS 2022 Accept_

### Official Review · Reviewer_4tXB · 2022-07-07

**Rating:** 6
**Confidence:** 3
**Soundness:** 4 excellent
**Presentation:** 4 excellent
**Contribution:** 3 good

**Summary:**

This paper observes that the training dynamics of deep neural networks is unstable in the sense that a tiny perturbation of the updates lead to significant changes in the dynamics (at the level of the trajectory), and also to a change in the final test accuracy (although rather benign). The authors propose to study this phenomenon from the point of view of numerical analysis and carry this program for a simplified model of a one-hidden-layer convnet with a single input and an infinite dimensional conv filter. For this toy model, a linearized analysis shows that there is a range of stable and unstable stepsizes and weight decay regularization. This observation is confirmed numerically on a deeper model.


**Questions:**

- In Section 5, I would rather call the system an (infinite dimensional) ODE rather than PDE because only a derivative in time is involved (while PDE is usually reserved for equations which involve partial derivatives in different variables).

**Strengths And Weaknesses:**

I enjoyed reading this paper: the presentation of the results is fair and with an appreciated scientific humbleness, the experiments are carried out with care and with a proper control of the various effects that could influence the outcome and the simple mathematical model is insightful.

The main weakness is that it is not clear whether this observation/analysis is an isolated remark or whether will lead to a fruitful line of research which will potentially lead to useful practical recommendations for deep learning (admittedly, this weakness applies to many papers that propose a new point of view). So the main question is whether the conclusions and recommandations remain relevant for more complex architectures (with more that one input, more than one filter, etc). In particular, it is not clear whether it is worth the computational cost (longer training time) to attempt to avoid this instability in practice by taking smaller step-sizes (because the variance on the test error is not that big).

---

> ### Author Response · Authors · 2022-08-02
> **Response to Reviewer 4**
>
> **(1)** *The main weakness is that it is not clear whether this observation/analysis is an isolated remark or whether will lead to a fruitful line of research which will potentially lead to useful practical recommendations for deep learning (admittedly, this weakness applies to many papers that propose a new point of view). So the main question is whether the conclusions and recommandations remain relevant for more complex architectures (with more that one input, more than one filter, etc). In particular, it is not clear whether it is worth the computational cost (longer training time) to attempt to avoid this instability in practice by taking smaller step-sizes (because the variance on the test error is not that big).*
>
> **->** We showed that empirically used practices such as learning rate decay schedules and weight regularization are not just helpful but fundamentally needed for stability. Moreover, as nets become more complicated in the future, having theoretical insight in how to optimize them not just effectively but as precisely as possible is likely to become more and more important. Please also see R2(2), instabilities arise locally; in larger networks, this means that instabilites would arise at one or more similarly structured sub-components as the network we analyzed, which was also verified empirically. We hope to design adaptive learning rates that choose small learning rates in localized regions where instabilities start, but otherwise larger rates away from these localized regions (see the discussion in the paper L316 - 319), preventing excessive computational time.
>
> **(Q1)** *In Section 5, I would rather call the system an (infinite dimensional) ODE rather than PDE because only a derivative in time is involved (while PDE is usually reserved for equations which involve partial derivatives in different variables).*
>
> **->** It's a relation of functions of time and space so we call it a PDE, whereas an ODE is only a relation of functions of one variable. We call it a PDE since there are multiple variables (i.e., time and a spatial variable). As such, even a derivative that is only in time is technically still a partial derivative rather than an ordinary derivative. That being said, it is also valid to think of it as an infinite family of ODEs and therefore we understand where the reviewer is coming from.

---

> > ### Comment · Reviewer_4tXB · 2022-08-08
> > **Thanks - Maintain opinion**
> >
> > Thanks for your answer. Ok, ODE vs PDE... it does not really matter.
> > I maintain my judgement that this paper would be an interesting contribution for the Neurips community and a "Technically solid, moderate-to-high impact paper".
> > I'm curious to see whether the other reviewers would update their opinion given the author's response?

---

### Official Review · Reviewer_rvJB · 2022-07-13

**Rating:** 5
**Confidence:** 3
**Soundness:** 3 good
**Presentation:** 3 good
**Contribution:** 2 fair

**Summary:**

This paper models optimization algorithms as PDEs and studies their stability to obtain bounds on the hyperparameters of the optimization algorithms (such as the learning rate). The authors first empirically demonstrate instabilities in training of deep networks induced by floating point approximations, and use that to make the case for choosing the learning rate carefully.

**Questions:**

The authors take a discrete time system (gradient descent optimization), and analyze a continuous analog of it (the gradient flow PDE), and then study ways to discretize the PDE. This is a curious approach - is there an advantage to this approach over studying the stability of the discrete time system in the first place?

**Limitations:**

Yes

**Strengths And Weaknesses:**

The paper is well supported by theory and experiments, and is easy to follow. Section 3 provides a nice introductory example to the technique that the authors are using. While the paper is technically sound in terms of the technique presented, there is no comparison to other bounds on the learning rate obtained from convex optimization, or BIBO stability of control systems. A more compelling case could be made by either showing that the PDE technique obtains bounds that are in line with previous observations, or that the technique obtains better bounds on the learning rate.

The paper would also be stronger if the authors could extend their approach to other optimization algorithms like Adam/Adagrad, etc.

---

> ### Author Response · Authors · 2022-08-02
> **Response to Reviewer 3**
>
> **(1)** *While the paper is technically sound in terms of the technique presented, there is no comparison to other bounds on the learning rate obtained from convex optimization, or BIBO stability of control systems. A more compelling case could be made by either showing that the PDE technique obtains bounds that are in line with previous observations, or that the technique obtains better bounds on the learning rate.*
>
> **->** Our analysis using Fourier analysis is analogous to analyzing stability of a control system which is based on analyzing the transfer function; the Fourier analysis does this on the imaginary axis, which showed instability. This is not a convex problem, we are not sure how much sense or where to begin in comparing it to convex optimization problems.
>
> **->** Our findings are consistent with common practices in deep learning. For example, we show regularization is not just helpful but necessary for strict convergence. Our analysis also explains why learning rate schedules are not just useful but necessary (as they eventually reach small values that lead to more stable schemes). The fact that stability is restrained and not unbounded also explains why learning rate schedules that don't obey the strict conditions of the theorem are still able to obtain usable results.
>
> **->** Beyond this we're unsure what previous theory we're trying reconcile, as most of the theories are developed under convexity and assumptions which barely apply to deep neural networks.
>
> **(2)** *The paper would also be stronger if the authors could extend their approach to other optimization algorithms like Adam/Adagrad, etc.*
>
> **->** We hope to do this in future work, but we think this study is a reasonable start.
>
> **(Q1)** *The authors take a discrete time system (gradient descent optimization), and analyze a continuous analog of it (the gradient flow PDE), and then study ways to discretize the PDE. This is a curious approach - is there an advantage to this approach over studying the stability of the discrete time system in the first place?*
>
> **->** This approach gives some insight into the discrete system as the dimensionality grows infinite, which shows fundamental properties remain beyond the discretization itself. The continuum is the limiting case of the discrete case, and thus meaningful to analyze analytically. While the approach is not yet widely adopted in the ML community, it is a common analysis in the PDE community and therefore it gives PDE people some insight and connection to deep neural networks.

---

### Official Review · Reviewer_ux6y · 2022-07-18

**Rating:** 6
**Confidence:** 3
**Soundness:** 3 good
**Presentation:** 3 good
**Contribution:** 2 fair

**Summary:**

In this paper, the authors show that there are numerical instabilities in the training of neural networks that inherently result from floating point errors. They empirically observe the phenomenon using a simple perturbation to the learning rate of SGD, which leads to some variations in the test performance. The authors further analyze one-layer CNN in which they can analytically derive an PDE of the gradient flow and use linear analysis to derive CFL conditions for stabilities.

**Questions:**

1) What does “restrained” mean exactly?

2) Theorem 5.2: What can we say about the PDE with RELU activation?

3) In Section 5, you consider the regularized cross-entropy loss. How about unregularized loss?


**Strengths And Weaknesses:**

This paper discovers some interesting empirical observations on the numerical instabilities of training neural networks with SGD. The author presents a theoretical analysis in a simplified setting and implications in choosing principle to choosing learning rates.

However, while there are variations in test accuracy shown in Table 1, I do not think they are that surprising. We do not know whether there is any other source of non-determinism in the implementation, which might cause the variations. I wonder if it is worth checking with a simple network on toy data using batch gradient descent?

On the theoretical side, the analysis, while revealing some conditions on the learning rate and weight decay, is rather limited to a simple architecture and the optimization requires regularization albeit common.

---

> ### Author Response · Authors · 2022-08-02
> **Response to Reviewer2**
>
> **(1)** *While there are variations in test accuracy shown in Table 1, I do not think they are that surprising. We do not know whether there is any other source of non-determinism in the implementation, which might cause the variations. I wonder if it is worth checking with a simple network on toy data using batch gradient descent?*
>
> **->** Yes we verified this; see Reviewer 1 Q1.
>
> **(2)** *On the theoretical side, the analysis, while revealing some conditions on the learning rate and weight decay, is rather limited to a simple architecture and the optimization requires regularization albeit common.*
>
> **->** See footnote 2 on Page 2 of the paper. Different from studies on "stability" that exist in deep learning literature, the numerical instabilities we are investigating arise locally (in the tensor space and iteration). If we showed it locally, then given a large network the instability would arise at one or more similarly structured sub-components. See also the dicsussion for Reviewer 1, Q3. So the analysis gives us insight to larger networks.
>
> **(Q1)** *What does “restrained” mean exactly?*
>
> **->** It means that the instabilites are not unbounded and are localized in the tensor space. They may drift from location to location but do not necessarily cause the weights to blow up. This is due to the nonlinearity of the network and the fact that the coefficients of the underlying linearization dynamical change, sometimes becoming more forgiving as an initial local instability begins to propagate and worsen.
>
> **(Q2)** *Theorem 5.2: What can we say about the PDE with RELU activation?*
>
> **->** Regarding the ReLU activation function, we have the following comments:
>
> **-> 1)** Initially, we intuitively conjectured that the instability that is exhibited in ReLU network is due to its discontinuity in the gradient at zero point (L146 - 150). However, the analysis on Swish shows the PDE is unstable NOT because of the discontinuity of the ReLU gradient; so we don't expect any conclusions of our theorems to change for the ReLU.
>
> **-> 2)** By removing non-differentiable point of ReLU, we were able to leverage PDE tools; which cannot be used for ReLU without further considerations. However, note that when $\beta \to + \infty$, ReLU becomes a special case of Swish. Thus the only thing we can say theoretically about ReLU is that since Swish retains the instability independent of Swish parameter, the result should continue to be unstable when $\beta \to + \infty$. This is consistent with our experiments for ReLU (large network) presented in Table 1 in the main paper.
>
> **(Q3)** *In Section 5, you consider the regularized cross-entropy loss. How about unregularized loss?*
>
> **->** The unregularized loss is the case when alpha = 0 in our theory. By our analysis (Table 3) it will always have restrained instabilities.

---

### Official Review · Reviewer_ojkU · 2022-07-19

**Rating:** 5
**Confidence:** 3
**Soundness:** 3 good
**Presentation:** 2 fair
**Contribution:** 2 fair

**Summary:**

The submission shows the numerical instabilities in the current training of deep networks (DNN) with SGD and claims that the numerical error is due to floating point arithmetic.
Then, it shows the condition of the learning rate and the weight decay to perform stable training from the perspective of numerical PDE.


**Questions:**

See the first point in Weaknesses.

**Limitations:**

Not related to the potential negative societal impact.

**Strengths And Weaknesses:**

Strengths
- The submission shows some evidence of numerical instabilities in reproducing the results of DNN training.
- The submission studies a simplified CNN by the numerical PDE analysis and offers some principles for choosing learning rate and weight decay.

Weaknesses
- The current experiment lacks some details, so the non-reproducibility of the training trajectory cannot be strictly guaranteed to come from floating point arithmetic. As we all know, if we perform GPU-based training, too many seeds need to be fixed (involving cuda, cudnn, etc.). Small negligence will lead to inconsistent trajectory and output in the training process. Therefore if it is claimed that the instability comes from the floating-point calculation, it should be ensured that the dynamic trajectories of weights can strictly coincide for several tries when k=1.
- The experimental findings and theoretical results seem to be separated. No matter what the unstable factors are, as long as the conditions given by the theorem are not satisfied, it appears that the optimal trajectory will diverge.
- The authors provide too many linear PDE-related validation experiments. Since the title and intro try to present the contribution from the perspective of DNN training, it is better to show the application of stability conditions in neural network training rather than the numerical simulation of PDE.
- The traditional convergence analysis for the optimizer can also get the conditions related to LR and weight. It's NOT that there is ``no theoretical result related to learning rate'', but it's not as good as the heuristic methods in practical use. Secondly, in practice, it seems that the large LR (violating the stability condition) can still be used to train DNN without the divergence described by the theorem. From this perspective, the conclusion of this submission is not very practical.

---

> ### Author Response · Authors · 2022-08-02
> **Response to Reviewer 1**
>
> **(1)** *The current experiment lacks some details, so the non-reproducibility of the training trajectory cannot be strictly guaranteed to come from floating point arithmetic. As we all know, if we perform GPU-based training, too many seeds need to be fixed (involving cuda, cudnn, etc.). Small negligence will lead to inconsistent trajectory and output in the training process. Therefore if it is claimed that the instability comes from the floating-point calculation, it should be ensured that the dynamic trajectories of weights can strictly coincide for several tries when k=1.*
>
> **->** We do fix all random seeds in our experiments and the floating points perturbation is the only randomness introduced. To verify that, as R1 suggested, we run experiments for multiple trials when k=1. Test accuracy results from each trial in the training process are in the following table. We also confirm that the dynamic trajectories of weights strictly coincide.
> |Trial\Epoch|0|20|40|60|80|100|120|140|160|180|200|
> |:-|:-|:-|:-|:-|:-|:-|:-|:-|:-|:-|:-|
> |1|33.84|79.22|90.9|91.19|93.07|93.33|93.38|93.47|93.44|93.39|93.47|
> |2|33.84|79.22|90.9|91.19|93.07|93.33|93.38|93.47|93.44|93.39|93.47|
> |3|33.84|79.22|90.9|91.19|93.07|93.33|93.38|93.47|93.44|93.39|93.47|
>
> **->** To further strengthen this point, we provide additional results on $k' = k\times 2^n$. Since base 2 (binary) is used in floating number (see L112 - 115 in the paper), multiplying or dividing by the power of 2 will not introduce floating point arithmetic perturbation, therefore for any integer $k$, results on $k'$ and $k$ will have the same trajectories. This is another piece of evidence showing that the instability comes ONLY from floating point error.  We also show that for each fixed odd $k$ final results are all consistent over different trials, as expected.  We also provided code in the supplementary (main4.py) for the reviewers to verify.
> |Trial|k=1|k=2|k=3|k=4|k=5|k=6|k=7|k=8|k=9|k=10|k=11|k=12|
> |:-|:-|:-|:-|:-|:-|:-|:-|:-|:-|:-|:-|:-|
> |1|93.47|93.47|93.29|93.47|93.40|93.29|93.72|93.47|93.62|93.40|93.79|93.29|
> |2|93.47|93.47|93.29|93.47|93.40|93.29|93.72|93.47|93.62|93.40|93.79|93.29|
> |3|93.47|93.47|93.29|93.47|93.40|93.29|93.72|93.47|93.62|93.40|93.79|93.29|
>
> **(2)** *The experimental findings and theoretical results seem to be separated. No matter what the unstable factors are, as long as the conditions given by the theorem are not satisfied, it appears that the optimal trajectory will diverge.*
>
> **->** It is highly probable that we have misunderstood the reviewer's intent here, because we completely agree with the conclusion (as we understood it) and see it as a positive point that goes to the heart of the paper rather than a criticism. Namely, the theorem puts forth necessary conditions for stability which therefore implies divergence whenever such conditions are not satisfied (regardless of the reason or factors). This is consistent with the experimental findings. We sincerely apologize if we have misunderstood the intended criticism, but we had trouble discerning any negative implication from this statement and therefore find ourselves unable to offer a fruitful response.
>
> **(3)** *The authors provide too many linear PDE-related validation experiments. Since the title and intro try to present the contribution from the perspective of DNN training, it is better to show the application of stability conditions in neural network training rather than the numerical simulation of PDE.*
>
> **->** The neural network structure can indeed be related to the discretization of a pde albeit a highly nonlinear one that exceeds the complexity of the simpler linear models considered in many of our experiments. Therefore, the reviewer's concern remains a reasonable one even through the related (though admittedly atypical) lens of numerical PDE analysis as compared to DNN training. However, stability analysis of nonlinear PDE discreteizations is almost always done through the analysis of local linearization. The reason for this is that most instabilities start locally and then propagate to more distance computational nodes (grid points in traditional PDE's or network weights in our comparative case), and such local stability can be related rigorously to the global stability of the corresponding linearized PDE. Of course, this linearization differs from place to place along the network and also changes dynamically as the optimization process proceeds, but this does not fundamentally change the structure of the underlying linear PDE, but just its coefficients, thereby allowing us to link the stability of the more complicated network (which relates to a more complicated nonlinear PDE) to the stability behavior of each of its constituent building blocks whose linearizations correspond to the linear PDE-related validation experiments.

---

> > ### Author Response · Authors · 2022-08-02
> > **Response to Reviewer 1 cont.**
> >
> > **(4)** *The traditional convergence analysis for the optimizer can also get the conditions related to LR and weight. It's NOT that there is ``no theoretical result related to learning rate'', but it's not as good as the heuristic methods in practical use. Secondly, in practice, it seems that the large LR (violating the stability condition) can still be used to train DNN without the divergence described by the theorem. From this perspective, the conclusion of this submission is not very practical.*
> >
> > **->** We are not sure where we claimed "no existing theoretical results related to learning rates", although we did point out that there was far more heuristics that seems to have outpaced the limited amount of theoretical work and understanding when it comes to stability. To that extend we agree with the reviewer's point that the heuristic methods have proven to be useful, powerful, and highly practical. Nevertheless we do see scientific merit and value in seeking to better understand heuristically observed behaviors from a theoretical perspective, especially for future scenarios as network structures evolve, or as training demands increase, and perhaps begin to exhibit new behaviors which don't immediately benefit from prior heuristic and empirical know-how. In fact, the reviewer's observation that in practice, we seem to get by quite successfully with large learning rates, violating the theorem, could even be considered a case in point. We we pointed out that the stabilities (due to the nonlinear nature of the network) are "restrained", due to the linearization coefficients changing over time, and therefore don't always lead to a numerical "blow-up" of coefficients. As such, we can essentially "over-drive" the strictly stable step size (learning rate) without a disaster ensuing, which has allowed this phenomenon to go undetected in the heuristic advancement of the art as it has not yet presented any practical consequences of note. However, it is not clear this will always be the case as demands and complexity evolve over time, and we therefore see value in beginning to better understand some of these theoretical properties even in current contexts which may not necessarily immediately benefit from these insights.

---

> > > ### Author Response · Authors · 2022-08-09
> > > **Reflection on point 2**
> > >
> > > We continued to reflect on the Reviewer’s point (2), and we may have understood the Reviewer's intent.  Perhaps by “diverge” the Reviewer meant a specific case of divergence where there is a full blowup of weights.  The experiments in Section 4 do not indicate a full blowup, which might have been where the Reviewer thought the experiments and theory seem separated.  If this is the case, please see our response to point (4) for a detailed discussion. Since the theory indicates instabilities are “restrained”, a full blowup is not to be expected in the experimental results of Section 4 rather significant divergence in the trajectories due to small perturbations.

---

> > > ### Comment · Reviewer_ojkU · 2022-08-09
> > > **Response to Author's rebuttal**
> > >
> > > Two of my most concerning issues are well responded to. Hence, I flip my attitude to positive.
> > >
> > > However, for point 3, I recommend more DNN-related results rather than the numerical simulation of PDE. I do not quite capture the relation between the response and my suggestion.

---

### Meta-Review · Area_Chair_ABFz · 2022-08-23

**Recommendation:** Accept
**Confidence:** Less certain

**Metareview:**

The paper studies the instabilities of neural networks. Most of the reviewers recommend acceptance. I also think the paper seems interesting.

**Award:**

No

---

### Decision · Program_Chairs · 2022-09-14

Accept